# The Progression of Symptoms in Post COVID-19 Patients: A Multicentre, Prospective, Observational Cohort Study

**DOI:** 10.3390/biomedicines12112493

**Published:** 2024-10-30

**Authors:** Merel E. B. Cornelissen, Myrthe M. Haarman, Jos W. R. Twisk, Laura Houweling, Nadia Baalbaki, Brigitte Sondermeijer, Rosanne J. H. C. G. Beijers, Debbie Gach, Lizan D. Bloemsma, Anke H. Maitland-van der Zee

**Affiliations:** 1Department of Pulmonary Medicine, Amsterdam UMC, University of Amsterdam, de Boelelaan 1117, 1081 HV Amsterdam, The Netherlandsa.h.maitland@amsterdamumc.nl (A.H.M.-v.d.Z.); 2Amsterdam Institute for Infection and Immunity, de Boelelaan 1117, 1081 HV Amsterdam, The Netherlands; 3Amsterdam Public Health, de Boelelaan 1117, 1081 HV Amsterdam, The Netherlands; 4Department of Epidemiology and Data Science, Amsterdam University Medical Centers, Van der Boechorststraat 7, 1081 BT Amsterdam, The Netherlands; 5Department of Environmental Epidemiology, Institute for Risk Assessment Sciences (IRAS), Utrecht University, Yalelaan 2, 3584 CM Utrecht, The Netherlands; 6Department of Pulmonology, Spaarne Hospital, Spaarnepoort 1, 2134 TM Hoofddorp, The Netherlands; 7Department of Respiratory Medicine, NUTRIM Institute of Nutrition and Translational Research in Metabolism, Maastricht University Medical Centre+, Universiteitssingel 50, 6229 ER Maastricht, The Netherlands

**Keywords:** post COVID-19 condition, symptom progression, trajectories, dyspnoea, fatigue

## Abstract

Background: Although the coronavirus disease 2019 (COVID-19) pandemic is no longer a public health emergency of international concern, 30% of COVID-19 patients still have long-term complaints. A better understanding of the progression of symptoms after COVID-19 is needed to reduce the burden of the post COVID-19 condition. Objective: This study aims to investigate the progression of symptoms, identify patterns of symptom progression, and assess their associations with patient characteristics. Methods: Within the P4O2 COVID-19 study, patients aged 40–65 years were recruited from five Dutch hospitals. At 3–6 and 12–18 months post COVID-19, medical data were collected, and pulmonary function tests were performed. In between, symptoms were assessed monthly with a questionnaire. Latent class mixed modelling was used to identify symptom progression patterns over time, with multinomial logistic regression to examine associations with patient characteristics. Results: Eighty-eight patients (aged 54.4 years, 48.9% males) were included. Three trajectories were identified for fatigue and dyspnoea: decreasing, high persistent, and low persistent. The odds of “decreasing fatigue” was higher for never smokers and participants in the lifestyle intervention and lower for those having a comorbidity. The odds of “decreasing dyspnoea” was higher for moderate COVID-19 patients and lifestyle intervention participants and lower for males, mild COVID-19 patients, and those with a higher age. Conclusions: Three distinct trajectories were identified for fatigue and dyspnoea, delineating patterns of symptom persistence following COVID-19. Sex, age, smoking status, participation in lifestyle interventions and COVID-19 severity were associated with the likelihood of belonging to different trajectories. These findings highlight the heterogeneity of the long-term symptoms experienced by post COVID-19 patients and emphasise the importance of personalised treatment strategies.

## 1. Introduction

Severe acute respiratory syndrome coronavirus 2 (SARS-CoV-2) has led to a major pandemic, resulting in substantial mortality and morbidity worldwide. Since the outbreak, more than 700 million coronavirus disease 2019 (COVID-19) cases and almost seven million deaths have been reported [1]. The average recovery time from acute COVID-19 is 2–3 weeks, depending on the symptom severity [2]. However, some cases may exhibit symptoms that are still present three months after the infection, last for at least two months and cannot be explained by an alternative diagnosis, referred to as the post COVID-19 condition by the World Health Organization (WHO) Delphi consensus definition [3]. Post COVID-19 condition has a significant social and economic impact. It leads to lower quality of life, and more people may need healthcare support in the near future, which could overburden the healthcare system [4,5].

Several studies have been conducted on symptoms associated with post COVID-19 condition. The results of recent systematic reviews [6,7,8] show that post COVID-19 symptoms are present in 30-45% of hospitalised and non-hospitalised ex-COVID-19 patients after four months [8]. These patients exhibit at least one post COVID-19 symptom for more than 30 days after the onset of infection, where fatigue and dyspnoea were the most prevalent [7]. Next to these symptoms, psychological symptoms were also frequently mentioned, like problems with concentration and sleeping, and difficulty finding words [9]. After one year, fatigue and dyspnoea were still reported by, respectively, 41% and 31% of the post COVID-19 patients [6], and, after two years, 30% of the patients still experienced symptoms [7]. It is still under debate in the literature whether the persistence of symptoms increases or decreases over time, as Ong et al. found a decrease in the number of patients with persistent symptoms from 30 to 90 days and a slight increase in this number from 90 to 180 days [10]. While previous studies have documented post-COVID-19 symptoms at specific time points, the progression of these symptoms over time remains largely unexplored.

Risk factors for the development of post COVID-19 condition and the persistence of symptoms are presented in several studies but are not always consistent. Frequently mentioned risk factors are age, sex, body mass index (BMI), smoking, comorbidities and the severity of acute COVID-19. According to these studies, women have an increased risk of developing post COVID-19 condition compared to men [11,12,13]. A higher BMI, smoking or former smoking and different comorbidities are also associated with a greater risk of post COVID-19 condition [13,14]. Frequently mentioned comorbidities are diabetes, hypertension, cardiovascular disease (CVD) and chronic obstructive pulmonary disease (COPD) [6,12,13,14]. Furthermore, age > 65 years and more severe acute COVID-19 increase the chance of having persistent symptoms of post COVID-19 condition [6,10,15,16,17].

A better understanding of the progression of symptoms after acute COVID-19 over time and its associations with patient characteristics are needed to create therapeutic strategies, to reduce the burden of post COVID-19 condition and to lower the social and economic impact. Therefore, the primary aim of this study is to investigate the progression of symptoms within the first twelve to eighteen months after acute SARS-CoV-2 infection in the Precision Medicine for more Oxygen (P4O2) COVID-19 cohort. The second aim is to identify trajectories of symptom progression for the most prevalent post COVID-19 symptoms. The third aim is to assess the associations between patient characteristics and these trajectories of symptom progression.

## 2. Materials and Methods

### 2.1. Study Design and Population

The P4O2 COVID-19 study is a multicentre, prospective, observational cohort study in the Netherlands and is approved by the ethical board of the Amsterdam University Medical Centre (UMC) (ref. no. NL74701.018.20). Details of the study design have been described by Baalbaki et al. [18]. In brief, 95 ex-COVID-19 patients were recruited between May 2021 and September 2022 at post COVID-19 outpatient clinics in five Dutch hospitals. Patients visited the post COVID-19 outpatient clinic 3–6 months after hospitalisation or, if not hospitalised, were referred to the outpatient clinic by their general practitioner three months after their positive polymerase chain reaction (PCR) test. Inclusion criteria were a confirmed SARS-CoV-2 infection (by PCR, serology tests or a COVID-19 Reporting and Data System (CO-RADS) score of 4 or 5), the ability to provide informed consent, aged 40–65 years, access to the internet and an understanding of the Dutch language. Exclusion criteria were the inability to provide informed consent, a terminal illness and participation in another study involving investigational or marketed products concomitantly or within four weeks prior to study entry or during the study. All participants were invited to two study visits: at 3–6 and 12–18 months after hospitalisation or positive PCR test. In between, patients completed monthly questionnaires at home and were invited to participate in a lifestyle intervention, consisting of personalised counselling on dietary quality and physical activity. Further detailed information of the lifestyle intervention is described by Baalbaki et al. [18]. The total duration of the study per participant was 9–12 months. 

### 2.2. Data Collection

Written informed consent was obtained during the first study visit, and baseline characteristics concerning the patient’s health status prior to and during COVID-19 were obtained from electronic patient files. During both study visits, pulmonary function tests (spirometry and diffusion capacity) were executed, and several questionnaires were administered. One of these questionnaires (Disease Progression and Adverse Events) contained questions on the frequency of pulmonary and extra-pulmonary symptoms (Appendix A). This questionnaire was developed in collaboration with post COVID-19 outpatient clinic physicians. Patients were asked to complete this questionnaire at home every month. This “monthly questionnaire” assessed whether a patient experienced different symptoms in the past two weeks and, if they did, how frequently they experienced them. The first monthly questionnaire was completed during the first study visit and the last questionnaire during the second study visit.

### 2.3. Outcome

The outcome of the study was the progression of symptoms after COVID-19. The symptoms that were assessed by the monthly questionnaire were fatigue, headache, chest pressure, dyspnoea, loss of smell and taste, abdominal pain, diarrhoea and obstipation. On the monthly questionnaire, patients were asked whether they had experienced each symptom (yes vs. no) in the past two weeks, and, if yes, whether this was daily or weekly. When the answer was “weekly”, the patient was subsequently asked how many times a week they experienced this symptom. We created a new variable to describe the weekly frequency (in days per week) that the patient experienced per symptom. When the patient’s answer was “daily”, the new variable was coded as “7”. When the answer was “weekly”, the patient subsequently stated how many times a week they experienced this symptom, and the new variable was given this number.

### 2.4. Patient Characteristics

In this study, the patient characteristics included age (in years), sex, BMI (in kg/m^2^), smoking status (current and ex-smoker vs. never smoker), COVID-19 severity (mild, moderate, or severe), diffusion capacity of carbon monoxide (DLCO) (in % predicted) and having a comorbidity (yes vs. no). COVID-19 severity was defined according to the WHO Clinical Progression Scale, based on oxygen supplementation during the infection [19]. Comorbidities that were assessed were COPD, asthma, interstitial lung disease (ILD), thrombosis, heart failure, renal failure, hepatic disease, diabetes, cancer, rheumatic diseases, CVD and neurological diseases.

### 2.5. Data Analysis

First, exploratory analyses were conducted. Patients who completed less than two monthly questionnaires were excluded from this study. Histograms were generated for each symptom to assess the mean weekly frequency at each time point. Fatigue and dyspnoea were selected for further trajectory analysis, since these symptoms were most frequently mentioned in the literature [6,7].

Second, to investigate patterns of symptom progression over time, latent class mixed modelling (LCMM) was performed (with R package LCMM) [20], which can divide the heterogeneity of symptom progression into more homogeneous trajectories of symptom progression. The model can handle missing data, meaning that it does not need the same number of measurements per time point or per participant. Two models were constructed: for the first model, fatigue was the dependent variable, and, for the second model, dyspnoea was the dependent variable. The time in months since acute infection was the independent variable for both models. Trajectories were added one by one to capture the underlying variability in symptom progression patterns. Model selection was based on the Akaike Information Criterion (AIC) and the Bayesian Information Criterion (BIC), with lower values indicating a better fit. Participants were divided over the trajectories based on probability scores, where the highest probability score indicated in which trajectory they belonged. Additionally, the best-fitting models were visually inspected by creating plots to ensure that the identified patterns aligned with our expectations of fatigue and dyspnoea progression. Furthermore, the assumption of normality of the residuals was checked.

Finally, to assess the associations between patient characteristics and the trajectories of symptom progression for fatigue and dyspnoea, first, univariable multinomial logistic regression analyses were performed for each patient characteristic separately; second, a multivariable multinomial logistic regression analysis was performed, including all significant patient characteristics (*p* < 0.05) from the univariable analyses.

Statistical analyses were performed using the R software, version 4.2.1.

## 3. Results

### 3.1. Baseline Characteristics

In total, 88 patients completed at least two monthly questionnaires and were included in this study. Their mean age was 54.4 ± 6.1 years and 48.9% were male (Table 1). The mean BMI was 30.5 ± 5.4 kg/m^2^ and 63.2% of the patients had at least one comorbidity. Of all patients, 88.6% were hospitalised, with a median duration of eight days, and most of the patients (63.6%) had moderate severity of COVID-19. There were 761 monthly questionnaires collected, with a median of 9.0 (25–75th percentile: 8.0–10.3) questionnaires per patient.

### 3.2. Symptom Progression

The mean weekly frequency (in days per week) of all eight symptoms was calculated for each month (Appendix A). The mean weekly frequency of fatigue declined from 5.2 days at four months to 3.6 days at eighteen months after the infection. Similarly, dyspnoea showed a decrease from 4.4 days at four months to 3.2 days at eighteen months after the infection. For chest pressure, the mean weekly frequency decreased from 1.8 days at four months to 0.1 days at eighteen months after the infection. The reported frequency of headache remained relatively stable throughout the study period, from 1.3 days per week at four months to 2.1 days at eighteen months after the infection. Loss of smell and taste, abdominal pain, diarrhoea and constipation showed consistently low scores throughout the study period.

### 3.3. Identified Trajectories of Fatigue

With LCMM, three trajectories were identified for fatigue (Figure 1). Trajectory one (“decreasing fatigue”, n = 23) contained patients whose weekly frequency decreased over time, trajectory two (“high persistent fatigue”, n = 39) consisted of patients whose weekly frequency remained high, and trajectory three (“low persistent fatigue”, n = 26) consisted of patients whose weekly frequency remained low over the study period. The AIC, BIC and probability scores per participant can be found in Appendix A. Differences in patient characteristics between the three trajectories are shown in Appendix A.

Multinomial logistic regression analyses, with “high persistent fatigue” as reference category, were performed for each patient characteristic separately (Appendix A). Thereafter, a multivariable analysis was performed with all significant patient characteristics (Table 2). The odds of “decreasing fatigue” was lower for patients with at least one comorbidity (odds ratio (OR) [95% confidence interval (CI)]: 0.54 [0.37, 0.80]), while it was higher for never smokers (1.90 [1.30, 2.78]) and participants in the lifestyle intervention (1.58 [1.07, 2.32]). The odds of “low persistent fatigue” was lower for mild COVID-19 patients (0.41 [0.18, 0.95]) and for patients with at least one comorbidity (0.63 [0.43, 0.93]), while it was higher for males (2.09 [1.38, 3.17]).

### 3.4. Identified Trajectories of Dyspnoea

Just like for fatigue, three trajectories were identified for dyspnoea (Figure 2): trajectory one (“decreasing dyspnoea”, n = 22), trajectory two (“high persistent dyspnoea”, n = 29) and trajectory three (“low persistent dyspnoea”, n = 37). The AIC, BIC and probability scores per participant can be found in Appendix A. Differences in patient characteristics between the three trajectories are shown in Appendix A.

For dyspnoea, the same univariable analyses were performed as for fatigue (Appendix A). When performing the multivariable analysis, the odds of “decreasing dyspnoea” was lower for patients with a higher age (OR [95% CI]: 0.88 [0.84, 0.92]), males (0.37 [0.22, 0.61]) and mild COVID-19 patients (0.12 [0.05, 0.31]), while it was higher for moderate COVID-19 patients (1.73 [1.05, 2.85]) and participants in the lifestyle intervention (2.21 [1.39, 3.53]) (Table 3). The odds of “low persistent dyspnoea” was lower for patients with a higher BMI (0.90 [0.86, 0.94]), patients with at least one comorbidity (0.56 [0.37, 0.84]) and mild COVID-19 patients (0.08 [0.03, 0.18]), while it was higher for males (1.66 [1.08, 2.55]) and patients with a higher DLCO % predicted (1.05 [1.04, 1.07]).

## 4. Discussion

This study showed an overall decline in the mean weekly frequency of fatigue, dyspnoea and chest pressure over the course of eighteen months after SARS-CoV-2 infection. Fatigue and dyspnoea demonstrated variability in symptom progression among three distinct trajectories: decreasing fatigue/dyspnoea, high persistent fatigue/dyspnoea, and low persistent fatigue/dyspnoea. Multinomial logistic regression analyses highlighted the significance of different patient characteristics in predicting the likelihood of belonging to different trajectories. The odds of “decreasing fatigue” was higher for never smokers and participants in the lifestyle intervention, while the odds of “low persistent fatigue” was higher for males. The odds of “decreasing dyspnoea” was higher for moderate COVID-19 patients and participants in the lifestyle intervention, and the odds of “low persistent dyspnoea” was higher for males and patients with a higher DLCO % predicted.

This study showed that the most reported persistent symptoms were fatigue, headache and dyspnoea, whereas diarrhoea, constipation and abdominal pain were less commonly reported. This is consistent with previous studies on symptoms after COVID-19, although these studies had a shorter follow-up period [21,22,23,24]. Brinkley et al. [25] found that the most reported symptoms by US adults were fatigue, headache, decreased smell and taste and cough. Symptoms that did not persist for a longer period were gastrointestinal symptoms (nausea, vomiting and diarrhoea). However, their follow-up period was only 28 days, whereas, in this study, symptoms were measured for between four and eighteen months. A longitudinal German study with a follow-up period of 12 months reported that the most likely symptoms to persist were reduced exercise capacity, fatigue and dyspnoea [9]. In our study, we found that approximately 40% of the patients experienced headache, with a weekly frequency ranging from 1.3 days to 2.1 days. In other studies, headache was found to be a persistent symptom in 18–22% of post COVID-19 patients [26,27,28]. In our study, the weekly frequencies of headache did not change considerably over time, so we were unable to create trajectories for this symptom. For future research, it could be of interest to study headache, since its prevalence is relatively high.

Factors associated with a high frequency of fatigue during the entire study period included female sex, (ex-)smoking, not being hospitalised, mild COVID-19 and the presence of comorbidities. The characteristics for a high frequency of dyspnoea were a higher age, female sex, a higher BMI, not being hospitalised, mild COVID-19 and a lower DLCO % predicted. These findings align with the existing literature reporting a higher age [14,21], higher BMI [25] and comorbidities [14,15] as risk factors for persistent symptoms after COVID-19. We found that females were more likely to experience persistent symptoms than males, which is in line with the studies by Bai et al. [29], who found female sex to be independently associated with a higher risk of post COVID-19 condition, and Onieva et al. [30], who found that females had a higher risk of persistent symptoms after infection. Furthermore, a study conducted in Suriname found female sex as a predictor of persistent symptoms at 3–4 months after COVID-19 [31]. Kamal et al. [15] also identified severe COVID-19 as a risk factor for the persistence of symptoms. This is in contrast with the results of this study, since mild COVID-19 was found to be associated with the persistence of symptoms. Notably, the study of Kamal et al. did not specify the time point of symptom assessment and used all symptoms together, instead of focussing on specific symptoms. When comparing the results of this study with those of a longitudinal study in a Dutch population [32], similar results were found. In their study, they found that slower recovery from dyspnoea was associated with a higher age, and slower recovery from fatigue was associated with having comorbidities.

Furthermore, a lifestyle intervention seems to influence fatigue and dyspnoea, since the odds of decreasing fatigue and decreasing dyspnoea were higher compared to high persistent fatigue and high persistent dyspnoea. This is in line with previous research, where it was found that a physically active lifestyle is important to overcome persistent symptoms and reduce the risk of post COVID-19 condition [33,34]. Detailed quantitative as well as qualitative analyses of the effects and experiences of participating in the lifestyle intervention are ongoing and will be further elaborated in future manuscripts.

Only one previous study was found that used trajectory analysis to find different patterns in persistent symptoms during the first year after SARS-CoV-2 infection [35]. This French cohort study, which used the Long COVID Symptom Tool to assess 53 symptoms, identified three trajectories, “highly persistent symptoms”, “rapidly decreasing symptoms” and “slowly decreasing symptoms”, and compared the patient characteristics between these trajectories. They also found that patients with highly persistent symptoms were older. However, contrary to the findings of this study, the authors did not observe associations with BMI or hospitalisation. This discrepancy might be because this study focused on the trajectories of fatigue and dyspnoea, whereas the French study created trajectories based on all symptoms together.

### Strengths and Limitations

This study has several strengths. First, the study had a prospective design, which offered the best overview of the symptoms that patients experienced at that moment. In addition, the questionnaire was administered every month, which provided longitudinal data and a detailed overview of the progression of symptoms over time. Second, in this study LCMM was used. This type of model can handle missing data points in longitudinal datasets and therefore prevented the exclusion of numerous cases that would have otherwise been eliminated due to incomplete data. Another strength is that the trajectories of fatigue and dyspnoea were created separately, instead of using all symptoms, so the symptoms could be treated more specifically.

This study also had some limitations. First, the questionnaire that was used to assess the frequency of symptoms was not a validated questionnaire. However, the questionnaire was developed in collaboration with post-COVID-19 outpatient clinic physicians, and the questions were very specific and easy to understand and interpret. Second, by only including patients who completed the monthly questionnaire at least twice, there was a risk of selection bias. Patients who completed fewer than two questionnaires might have been more likely to have experienced symptom resolution, or, on the contrary, patients experiencing a high disease burden may have been unable to complete the questionnaires. Third, there was no information on the health status prior to COVID-19, which made it challenging to obtain conclusions on whether the reported symptoms were caused by the SARS-CoV-2 infection. Although the study included information on comorbidities, which provided some insight into participants’ health profiles, a comprehensive understanding of their pre-existing conditions would have strengthened the interpretation of symptom presentation. Fourth, there was no control group (i.e., persons without post COVID-19 condition), which made it difficult to determine whether the observed symptoms were specific to post COVID-19 condition or could be attributed to other factors. Including a control group would have enhanced the study’s validity and enabled a more nuanced understanding of post COVID-19 condition’s symptomatology. Lastly, the sample size was small, which makes it difficult to generalise the findings to all post COVID-19 patients. Increasing the sample size in future studies would enhance the reliability, validity and generalisability of these findings.

## 5. Conclusions

Three distinct trajectories were identified for fatigue and dyspnoea, delineating patterns of symptom persistence in post COVID-19 patients. Sex, age, smoking status, participation in a lifestyle intervention and COVID-19 severity were associated with the likelihood of belonging to different trajectories. These findings highlight the heterogeneity of the long-term symptoms experienced by post COVID-19 patients and emphasise the importance of personalised treatment strategies. Future research should focus on validating these findings in larger cohorts with a longer follow-up time and exploring treatments targeting modifiable risk factors to improve the long-term outcomes in COVID-19 survivors.

## Figures and Tables

**Figure 1 biomedicines-12-02493-f001:**
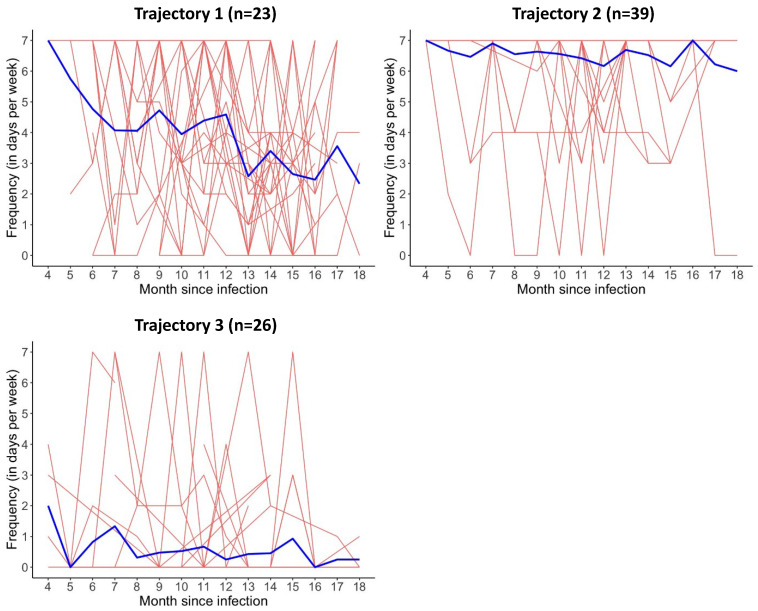
Three trajectories based on the weekly frequency of fatigue from month 4 to month 18 after SARS-CoV-2 infection. Each red line represents a patient and each blue line represents a trajectory.

**Figure 2 biomedicines-12-02493-f002:**
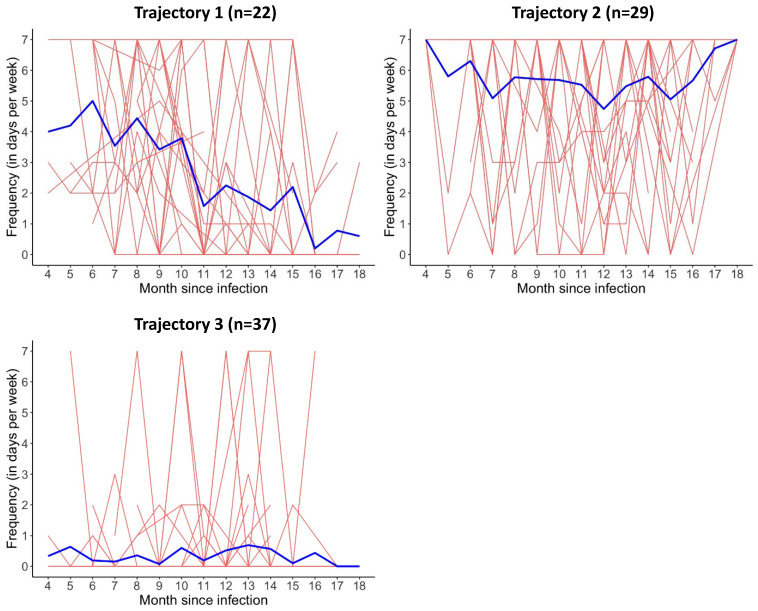
Three trajectories based on the weekly frequency of dyspnoea from month 4 to month 18 after SARS-CoV-2 infection. Each red line represents a patient and each blue line represents a trajectory.

**Table 1 biomedicines-12-02493-t001:** Patient characteristics (n = 88).

	Mean ± SD, Median (25th–75th Percentile) or n (%)
Age, in years	54.4 ± 6.1
Male sex	44 (48.9)
Ethnicity	
Caucasian	67/84 (79.8)
Other	17/84 (20.2)
BMI, in kg/m^2^	30.5 ± 5.4
Smoking status	
Current	4 (4.6)
Ex	47 (53.4)
Never	37 (42.1)
At least one comorbidity ^a^	55/87 (63.2)
Comorbidities	
Heart failure	5/87 (5.8)
Renal failure	6/85 (7.1)
Diabetes	13/87 (14.9)
COPD	6/87 (6.9)
Asthma	16/87 (18.4)
Cardiovascular disease	22/86 (25.6)
Hospitalised	78 (88.6)
Hospital duration, in days	8.0 (4.0–15.5)
COVID-19 severity ^b^	
Mild	10 (11.4)
Moderate	56 (63.6)
Severe	22 (25.0)
Number of monthly questionnaires completed	9.0 (8.0–10.3)
Participated in lifestyle intervention	42 (47.7)

BMI: body mass index, COPD: chronic obstructive pulmonary disease, DLCO: diffusion capacity for carbon monoxide, SD: standard deviation. ^a^ At least one of the following comorbidities: COPD, asthma, interstitial lung disease, thrombosis, heart failure, renal failure, hepatic disease, diabetes, cancer, rheumatic disease, CVD and neurological disease. ^b^ According to the WHO Clinical Progression Scale.

**Table 2 biomedicines-12-02493-t002:** Multivariable associations between patient characteristics and the different trajectories for fatigue.

	OR (95% CI) for Belonging to Trajectory 1 “Decreasing Fatigue” (n = 23)	OR (95% CI) for Belonging to Trajectory 3 “Low Persistent Fatigue” (n = 26)
Age ^a^	0.96 (0.93, 1.00)	1.03 (0.99, 1.06)
Male sex	0.73 (0.48, 1.11)	**2.09 (1.38, 3.17)**
BMI ^a^	0.96 (0.92, 1.00)	0.98 (0.94, 1.01)
Never smoker ^b^	**1.90 (1.30, 2.78)**	1.19 (0.81, 1.75)
Moderate COVID-19 ^c^	1.38 (0.87, 2.17)	0.91 (0.60, 1.36)
Mild COVID-19 ^c^	0.55 (0.26, 1.17)	**0.41 (0.18, 0.95)**
At least one comorbidity ^d^	**0.54 (0.37, 0.80)**	**0.63 (0.43, 0.93)**
Lifestyle intervention	**1.58 (1.07, 2.32)**	0.98 (0.67, 1.43)

BMI: body mass index, CI: confidence interval, OR: odds ratio. ORs were obtained by comparing the trajectories with the reference category trajectory 2 “high persistent fatigue”. The significant ORs are marked in bold. ^a^ OR is per unit increase in age (years) or BMI (kg/m^2^). ^b^ Reference category is current and ex-smoker. ^c^ According to the WHO Clinical Progression Scale. Reference category is severe COVID-19. ^d^ At least one of the following comorbidities: COPD, asthma, interstitial lung disease, thrombosis, heart failure, renal failure, hepatic disease, diabetes, cancer, rheumatic disease, CVD and neurological disease.

**Table 3 biomedicines-12-02493-t003:** Multivariable associations between patient characteristics and the different trajectories for dyspnoea.

	OR (95% CI) for Belonging to Trajectory 1 “Decreasing Dyspnoea” (n = 22)	OR (95% CI) for Belonging to Trajectory 3 “Low Persistent Dyspnoea” (n = 37)
Age ^a^	**0.88 (0.84, 0.92)**	0.97 (0.94, 1.01)
Male sex	**0.37 (0.22, 0.61)**	**1.66 (1.08, 2.55)**
BMI ^a^	0.98 (0.94, 1.02)	**0.90 (0.86, 0.94)**
Moderate COVID-19 ^b^	**1.73 (1.05, 2.85)**	1.17 (0.75, 1.81)
Mild COVID-19 ^b^	**0.12 (0.05, 0.31)**	**0.08 (0.03, 0.18)**
DLCO ^a^	1.00 (0.99, 1.02)	**1.05 (1.04, 1.07)**
At least 1 comorbidity ^c^	1.42 (0.88, 2.28)	**0.56 (0.37, 0.84)**
Lifestyle intervention	**2.21 (1.39, 3.53)**	0.77 (0.51, 1.14)

BMI: body mass index, CI: confidence interval, OR: odds ratio. ORs were obtained by comparing the trajectories with the reference category trajectory 2 “high persistent dyspnoea”. The significant ORs are marked in bold. ^a^ OR is per unit increase in age (years), BMI (kg/m^2^) or DLCO (% predicted). ^b^ According to the WHO Clinical Progression Scale. Reference category is severe COVID-19. ^c^ At least one of the following comorbidities: COPD, asthma, interstitial lung disease, thrombosis, heart failure, renal failure, hepatic disease, diabetes, cancer, rheumatic disease, CVD and neurological disease.

## Data Availability

The datasets generated and/or analysed during the current study are not publicly available due to agreements made by the consortium, which only allow access by each consortium partner to specific data that answers their pre-specified research questions, but they are available from the corresponding author on reasonable request. A request for access to the data by organisations outside of the consortium can be submitted to the P4O2 Data Committee (via p4o2@amsterdamumc.nl) and the research will need to be performed in collaboration with one of the P4O2 consortium partners.

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
