# Peer review of "The Progression of Symptoms in Post COVID-19 Patients: A Multicentre, Prospective, Observational Cohort Study"

_biomedicines, 2024, doi:10.3390/biomedicines12112493_

Round 1

Reviewer 1 Report

Comments and Suggestions for Authors

The article “Progression of Symptoms in Post COVID-19 Patients: A 2 Multicentre, Prospective, Observational Cohort Study” is written well and is very informative about symptoms occurred post COVID-19 Patients. However, there are some minor comments

In abstract authors should write the importance of this study, as the covid is diminished now.  How it will benefit the physicians and patients.

In material methods section: duration of survey/study is not mentioned.  

What are the different tools used to collect the data?

Author Response

We thank the reviewer for his/her comments and suggestions. We have adjusted the manuscript accordingly and responses to all comments have been provided below.

Comment 1: In abstract authors should write the importance of this study, as the covid is diminished now.  How it will benefit the physicians and patients.

Response 1: We agree with this comment and added a sentence about post COVID-19 in lines 20-22 at page 1. “Even though the coronavirus disease 2019 (COVID-19) pandemic is no longer a public health emergency of international concern, 30% of COVID-19 patients still have long-term complaints.

Comment 2: In material methods section: duration of survey/study is not mentioned. 

Response 2: Although we mentioned that there were two study visits performed at 3-6 months and at 12-18 months after infection, we agree that we could be more clear about the study duration. Therefore, we added a sentence about the study duration in line 110-111 at page 3. “The total duration of the study per participant was 9-12 months.

Comment 3: What are the different tools used to collect the data?

Response 3: As detailed in section 2.2 Data collection (lines 111-123 at page 3) the following tools were used for data collection in this study: questionnaires, medical records and pulmonary function tests.

Reviewer 2 Report

Comments and Suggestions for Authors

Dear Authors, I congratulate you for the manuscript submitted entitled "The progression of symptoms in post COVID-19 patients: a multicentre, prospective, observational cohort study", which addresses an important issue like symptomatological persistence associtated to this disease and whether some factors, like age, gender, smoking status, among others, may influence its progression. Overall the text follows a coherent line of thought and the updated literature that grounds your introduction translates your knowledge on the topic addressed.

However, there were some minor issues I woud like you to reflect on, which I believe could contribute to improve your work.

Keep the good work!

Regards  

Author Response

We thank the reviewer for his/her comments and suggestions. We have adjusted the manuscript accordingly and responses to all comments have been provided below.

Comment 1: (lines 60-61): "However, the number of patients with persistent symptoms seems to decline over time [10]." Actually, the persistency evolution of COVID-19 symptoms in the Ong and colleagues' study was not so linear as this sentence seems to convey, as there was an increase of one patient with persistent symptoms of COVID-19 at 90 to 180 days of post symptoms onset. Besides, the same study found that age, ethnicity and severity of acute infection were associated with increased likelihood of persistent symptoms.

Response 1: In response to this comment, we have added more results from the study by Ong and colleagues to the introduction in lines 65-68 and lines 79-81 at page 2. “It is still under debate in literature whether the persistence of symptoms increases or decreases over time as Ong et al. found a decrease in number of patients with persistent symptoms from 30 to 90 days and a slight increase in number from 90 to 180 days [10].” and “Furthermore, age > 65 years and more severe acute COVID-19 increase the chance of having persistent symptoms of post COVID-19 condition [6, 10, 15-17].”

Comment 2: (lines 107-116): "During both study visits, pulmonary function tests (spirometry and diffusion capacity) were executed, and questionnaires were administered. One of the questionnaires (Disease progression and adverse events) contained questions on the frequency of pulmonary and extra-pulmonary symptoms (Supplementary material 1). This questionnaire was developed in collaboration with post-COVID-19 outpatient clinic physicians. Patients were also asked to complete the questionnaire at home every month. This “monthly questionnaire” assessed whether a patient experienced different symptoms in the past two weeks and if they did, how frequently they experienced it. The first monthly questionnaire was completed during the first study visit and the last questionnaire during the second study visit." How many different questionnaires did your sample fulfil and at which timepoints did they do it? The way you have written - especially when you say ".... questionnaires were administered. One of the questionnaires..." - may raise the doubt different questionnaires had been also applied. It is a semantic interpretation issue, so I suggest this passage should be rephrased to dispel any doubts about the data collection instrument used in your research.

Response 2: We tried to be more clear about the questionnaires that were administered during the study visits and the questionnaire that was used every month in between the study visits. We rewrote some sentences in this part (lines 115-124 at page 3). “During both study visits, pulmonary function tests (spirometry and diffusion capacity) were executed, and several questionnaires were administered. One of these questionnaires (Disease progression and adverse events) contained questions on the frequency of pulmonary and extra-pulmonary symptoms (Supplementary material 1). This questionnaire was developed in collaboration with post-COVID-19 outpatient clinic physicians. Patients were asked to complete this questionnaire at home every month. This “monthly questionnaire” assessed whether a patient experienced different symptoms in the past two weeks and if they did, how frequently they experienced it. The first monthly questionnaire was completed during the first study visit and the last questionnaire during the second study visit.

Comment 3: (lines 118-127): "The outcome was the progression of symptoms after COVID-19. The symptoms that were assessed by the monthly questionnaire were: fatigue, headache, chest pressure, dyspnoea, loss of smell and taste, abdominal pain, diarrhoea, and obstipation. A new variable was created to describe the weekly frequency (in days per week) that the patient experienced a specific symptom. In the monthly questionnaire, patients were asked whether they have experienced each symptom (yes vs. no) in the past two weeks, and if yes, whether this was daily or weekly. When the patient’s answer was ‘daily’, the new variable was coded as ‘7’. When the answer was ‘weekly’, the patient was subsequently asked how many times a week they experienced this symptom and the new variable was given this number." When was that weekly frequency variable created? Depending on your answer, it may not be worthy to justify its creation, unless it has stemmed from a need realized after a trial-test, which does not seem to be the case. Besides, the explanation on how the questionnaire should be fulfilled by the patients should not be in this section but in the previous. In outcomes, you should describe how your data once collected through the questionnaire was coded and organized, in order to answer your research questions. For instance, for each patient, a pool of data (e.g., age, sex, smoking habits, BMI, COVID-19 severity, comorbidities, etc.) was tabulated with the respective symptomatological data, as how often general practitioner and/or a specialist at the hospital was visited in the last month, as well as how often symptoms like fatigue, headache, chest pressure, dyspnoea, loss of smell and/or taste, abdominal pain, diarrhoea, or obstipation were felt or not in the past two weeks.

Response 3: Thank you for this comment. We agree that we may have described it a bit unclear. What we meant to describe was how we coded and organized the collected data. We have changed this paragraph as follows (lines 129-139 at page 3). “In the monthly questionnaire, patients were asked whether they have experienced each symptom (yes vs. no) in the past two weeks, and if yes, whether this was daily or weekly. When the answer was ‘weekly’, the patient was subsequently asked how many times a week they experienced this symptom. We created a new variable to describe the weekly frequency (in days per week) that the patient experienced per symptom. When the patient’s answer was ‘daily’, the new variable was coded as ‘7’. When the answer was ‘weekly’ and the patient subsequently answered how many times a week they experienced this symptom, the new variable was given this number.

Comment 4: (lines 185-187): "The reported frequency of headache remained relatively stable throughout the study period from 1.3 days per week at four months to 2.1 days at eighteen months after infection.". This increase of headache frequency the fourth and the eighteenth month after infection, which goes against the trend observed in the other symptoms, did not raise enough interest to be further discussed? I have to express my surprise, as headache has been found to be the most common neurological symptoms of the COVID_19 (Caronna et al., 2020, 2021; Rocha-Filho, 2022; Tana et al., 2022) and that coexisting comorbidity (e.g., cerebrovascular disease, psychiatric disorders), and psychosocial stressors have been associated specifically to persistent headache (Pilotto et al., 2021; Magdy et al., 2022).

Response 4: We do agree that headache might have been interesting to look further into. However, in our study the weekly frequencies of headache were relatively low and did not change considerably over time (ranging from 1.3 to 2.1 days). We therefore were not able to make trajectories of this symptom. In respons to this comment, we extended a bit more on headache (and comparison with previous literature) in the discussion in lines 279-284 at page 8. “In our study we found approximately 40% of the patients experienced headache with a weekly frequency ranging from 1.3 days to 2.1 days. In other studies, headache is found to be a persistent symptom in 18-22% of post COVID-19 patients [26-28]. In our study the weekly frequencies of headache did not change considerably over time, so we were unable to make trajectories for this symptom. For future research it could be of interest to study headache, since the prevalence is relatively high.

Comment 5: (lines 250-251): "Multinomial logistic regression analyses highlighted the significance of several patient characteristics in predicting the likelihood of belonging to different trajectories.". I'd change "several" by "different", as despite 5 characteristics have been found to be somehow associated to trajectories of symptoms progression, no more than 2 characteristics were associated to each of three trajectories settled by the latent class mixed modelling applied, which induce readers to misinterpret the likelihood prediction, even because there were characteristics associated to more than you trajectory, like gender and lifestyle intervention, thus reducing the variability associated to the term "several".

Response 5: We replaced “several” by “different” in line 263 at page 8. “Multinomial logistic regression analyses highlighted the significance of different patient characteristics in predicting the likelihood of belonging to different trajectories.

Comment 6: (lines 253-254/255-256): "... the odds of “low persistent fatigue” was higher for males."/ "... the odds of “low persistent dyspnoea” was higher for males ...". This low persistency favourable to men is supported by recent research that have found long COVID seems to follow a pattern which points to the adult female gender as the group most affected by this disease and exhibit a higher risk of persistent symptoms and display distinct patterns in symptom clusters and functional status compared to males (Bai et al., 2022; Gorenshtein et al., 2024; Marcilla-Toribio et al., 2024; Onieva et al., 2024). This issue should be better discussed on your research.

Response 6: We agree that sex has a big influence on the persistence of symptoms in post COVID-19 patients. In response to this comment, we mentioned the sex differences found in other studies in lines 291-294 at page 8. “We found that females were more likely to experience persistent symptoms than males, which is in line with studies from Bai et al. [29] who found female sex to be independently associated with a higher risk of post COVID-19 condition and Onieva et al. [30] who found that females have a higher risk of persistent symptoms after infection.

Comment 7: (lines 260-262): "Brinkley et al. [25] found that most reported symptoms by US adults were fatigue, headache, decreased smell and taste, and cough.". This passage could be the trigger to discuss the increasing of the weekly headache episodes commented on lines 185-187.

Response 7: We added more literature on headache after SARS-CoV-2 infection in the discussion part of the manuscript. Please see our response to comment 4.  

Comment 8: (lines 284-288): "...lifestyle intervention seems to influence fatigue and dyspnoea, since the odds of decreasing fatigue and decreasing dyspnoea were higher compared to high persistent fatigue and high persistent dyspnoea. Detailed quantitative as well as quality analysis on the effects and experiences of participating in the lifestyle intervention are ongoing and will be further elaborated in future manuscripts.". Although dyspnoea have not been objectively studied in most of the research published, there are in turn an extensive pool of evidence that found active and healthy lifestyles to be an effective way to mitigate the longterm effects of COVID-19 infection (Abdelelgawad et al., 2024; Centorbi et al., 2024; Coscia et al., 2023; Sun et al., 2024; ...)

Response 8: In response to this comment, we have added some literature in lines 307-309 at page 9. “This is in line with previous research where it was found that a physical active lifestyle is important to overcome persistent symptoms and reduce the risk of post COVID-19 condition [33, 34].”

Reviewer 3 Report

Comments and Suggestions for Authors

Review Report for “The Progression of Symptoms in Post COVID-19 Patients: A 2 Multicentre, Prospective, Observational Cohort Study”

Dear Editor

Thank you for opportunity to review this paper which has been submitted by Merel E. B. Cornelissen on behalf of the P4O2 COVID-19 consortium-it is noteworthy that P4O2 COVID-19 study is a multicentre, prospective, observational cohort study in the Netherlands.

There are many rationales to conduct such a study, because a better understanding of the progression of symptoms after acute COVID-19 over time and its associations with patient characteristics are needed to create therapeutic strategies, in order to reduce the burden of post COVID-19 condition and to lower the social and economic impact. The main aim of this study was to expolre the progression of symptoms within the first twelve to eighteen months after acute SARS-CoV-2 infection. The second aim was to to identify trajectories of symptom progression for the most prevalent post COVID-19 81 symptoms. The third aim was to assess the associations between patient characteristics and these trajectories of symptom progression.

Strengths:

Interdiction

This section covers the current literature and highlights the gaps and really written well. Aims and question of the study are presented clearly.

Methods:

This section enjoys form a standard methodology, acceptable sample size, and proper statistical analyses.

Results:

Results are presented clearly. I checked the data and their analysis, which were all sound and perfect with no flaw. Results are novel and considerable in terms of usual medical practice and personalized medicine. This section definitely adds to the literature and our understanding of post-covid-19 symptoms.

Discussion

This section is also covers enough literature review; authors have sufficiently compared their findings with previously published materials and have discussed the finding in an efficient way. Authors conclude that 3 distinct trajectories can be identified for fatigue and dyspnoea, delineating patterns of symptom persistence in post COVID-19 patients. Sex, age, smoking status, participation in the lifestyle intervention and COVID-19 severity were associated with the likelihood of belonging to different trajectories. These results highlight the heterogeneity of long-term symptoms experienced by post COVID-19 patients and emphasize the importance of personalized treatment strategies. Authors have also proposed some suggestions for future research should including focusing on validating these findings in larger cohorts with a longer follow-up time and exploring treatments targeting modifiable risk factors to improve long-term outcomes in COVID-19 survivors.  

More importantly authors have included sections for Strengths and limitation of their study which can effectively help future research.

I am not a native English speaker and am not qualified to present my opinion, but the paper reads quite well and flerntly. I can endorse this paper for publication as it is, with no more comments.  Thank you very much again for this opportunity.

Author Response

We would like to thank the reviewer for reviewing the manuscript. 

Reviewer 4 Report

Comments and Suggestions for Authors

The manuscript titled "The Progression of Symptoms in Post COVID-19 Patients: A Multicentre, Prospective, Observational Cohort Study" examines the progression of post-COVID-19 symptoms such as fatigue and dyspnoea in patients aged 40-65, over 12 to 18 months. The study, conducted across five Dutch hospitals, identified three symptom progression trajectories: decreasing, high persistent, and low persistent. Factors such as sex, age, smoking status, COVID-19 severity, and participation in a lifestyle intervention were found to influence these trajectories. The study highlights the need for personalized treatment strategies to address the varied experiences of post-COVID-19 patients.

Overall, the manuscript is methodologically sound and contributes significantly to understanding post-COVID-19 conditions. As the authors acknowledged,  further studies with larger cohorts and validated tools would be beneficial to confirm these findings.

Author Response

(The authors gave the same response as above.)
